# The Impact of National Activities on Antibiotic Consumption in Hospitals and Different Departments over a 14-Year Period

**DOI:** 10.3390/antibiotics13060498

**Published:** 2024-05-28

**Authors:** Milan Čižman, Tamara Kastrin, Bojana Beović, Aleksander Mahnič, Tom Bajec

**Affiliations:** 1Department of Infectious Diseases, University Medical Centre, 1000 Ljubljana, Slovenia; bojana.beovic@kclj.si; 2Department for Public Health Microbiology, National Laboratory of Health, Environment and Food, 1000 Ljubljana, Slovenia; 3Faculty of Medicine, University of Ljubljana, 1000 Ljubljana, Slovenia; 4Department for Microbiological Research, National Laboratory of Health, Environment and Food, 2000 Maribor, Slovenia; aleksander.mahnic@nlzoh.si; 5Tomtim d.o.o, 1000 Ljubljana, Slovenia; tbajec@siol.net

**Keywords:** antibiotics, consumption, surveillance, Slovenia

## Abstract

The aim of this study was to assess the use of antibiotics in hospitals and different departments over 14 years (2006–2019) and the impact of various national activities related to this, including national audits of the use of antibiotics for systemic use. The consumption of antibiotics for systemic use (J01) from all Slovenian hospitals (n = 29) and five departments (internal medicine, surgery, ICU (medicine, surgery), paediatrics and gynaecology/obstetrics) was collected. Total hospital consumption was expressed as the number of defined daily doses (DDDs) per 1000 inhabitants per day (DID), the number of DDDs/100 bed days and the number of DDDs/100 admissions. Over 14 years, J01 hospital consumption increased by 13.8%, expressed in DDDs/100 bed days (*p* = 0.002). In 2019, compared to 2006, the consumption of J01, expressed in DDD/100 bed days, increased from 19.9% to 33.1% in all departments, except intensive care units. J01 consumption expressed in DDD/100 admissions increased by 7.0% to 39.4% in all but paediatric wards (where it decreased by 12.7%). In all years, we observed large variations in the consumption of antibiotics in departments of the same type. The effectiveness of audit interventions aimed at optimizing antibiotic consumption exhibited notable variation across hospitals, with specialized facilities generally demonstrating superior outcomes compared to general hospitals.

## 1. Introduction

Antimicrobial agents, predominantly antibiotics (antibacterials), are commonly prescribed in ambulatory care, hospital care and long-term care. Data from the European Centre for Disease Prevention and Control (ECDC) show that 20–60% of inpatients in acute care hospitals are treated with antibiotics, with even higher percentages in intensive care units (ICUs) and haematology–oncology departments [1,2]. According to data collected by the Organization for Economic Co-operation and Development (OECD), inappropriate use of antibiotics may amount to up to 50% of all antimicrobials in human healthcare and may be as high as 90% in long-term care facilities and ambulatory care [3]. Several studies have demonstrated the link between antibiotic consumption and the emergence of infections with antibiotic-resistant pathogens, which underlines the importance of responsible use of antibiotics [4,5]. Surveillance of antimicrobial consumption is one of the action points of the European Commission’s “One Health Action against Antimicrobial Resistance” [6].

To cope with the problem of antibiotic resistance, both national and international bodies have established systems to monitor and compare antibiotic consumption [7,8,9,10]. However, most of the studies published to date only present antibiotic consumption data at the national level and in hospitals in general [11,12,13,14], with only a limited number of studies describing department-level consumption [15]. In order to plan interventions to improve antibiotic use, we need antibiotic consumption data at the department level. To the best of our knowledge, there are no long-term national data from individual departments in the current literature.

Thus, the aim of this study was to evaluate antibiotic consumption in hospitals and different departments during a 14-year period and to examine the impact of national activities related to this, including national audits on the consumption of antibiotics for systemic use. 

## 2. Results

### 2.1. Consumption of Antibiotics for Systemic Use (J01) in Hospitals

The consumption of J01 in all Slovenian hospitals is shown in Table 1. In the observed period, the number of admissions in hospitals increased by 6.9%, while the number of bed days and average length of stay (LTS) decreased by 3.9% and 15.5%, respectively. Consumption of J01 expressed in DIDs did not change significantly in the observed 14-year period (*p* = 0.570, Figure 1). Consumption of J01 expressed in DDDs/100 bed days increased significantly (*p* = 0.002, Figure 1) in the observed period, while on the other hand, consumption expressed in DDDs/100 admissions showed a borderline statistically significant decrease (*p* = 0.097, Figure 1).

### 2.2. Consumption of Antibiotics for Systemic Use (J01) in Different Departments

In 2019 compared to 2006, the consumption of J01 expressed in DDDs/100 bed days increased in all departments, except in both ICUs, from 19.9% (Internal Medicine) to 33.1% (Gynaecology/Obstetrics). In medical ICUs the consumption decreased by 7.1% and in surgical ICUs by 5.4%, respectively. In both years, we can see large variations in antibiotic consumption in departments of the same type. In 2006, the variations were on average 3.05 (ranging from 1.7 to 5.0), and in 2019 they were on average 2.2 (ranging from 1.6 to 3.0), respectively. The ratio between consumption in medical ICUs and medical departments was 2.5 in 2006 and 1.9 in 2019. The ratio between consumption in surgical ICUs and surgical departments was 3.5 in 2006 and 2.6 in 2019.

Consumption of J01 expressed in DDDs/100 admissions increased by 7.0% (Medical ICU) to 39.4% (Surgery) in 2019 compared with the year 2006 in all cases except in paediatric departments, where it decreased by 12.7%. Large variations in consumption expressed in DDDs/100 admissions were also found in departments of the same type. The mean variation in 2019 was 3.2 (ranging from 2.1 to 4.5), while in 2006 it was 4.0 (ranging from 2.2 to 6.4). The ratio between the consumption of J01 expressed in DDDs/100 admissions in medical ICUs and medical departments was 2.7 and 3.8 in 2006 and 2019, respectively, and 1.1 and 3.1 in surgical departments, respectively.

All significant trends consistently point towards an increase in antibiotic consumption during the observed period, as evident in both DDDs/100 bed days and DDDs/100 admissions for the investigated departments (Figure 2). In all departments, we observed an increased number of admissions from 3.1% in gynaecology/obstetrics to 81.2% in paediatric departments. In internal medicine departments, the number of admissions increased by 59,7%%, followed by surgery (32,4%), surgical ICU (30,2%) and medical ICU (7.0%). At the same time, the average LOS also decreased in all departments from −2.7% in medical ICU to 70,1% in paediatrics. In gynaecology/obstetrics, internal medicine, surgery and surgical ICU, the average LOS decreased by 43%, 38,9%, 36.1% and 14%, respectively. The number of bed days increased in surgical (12%) and medical ICU (2.9%) and decreased in other departments from internal medicine (2.6%) to genecology/obstetrics (41.1%). Department consumption expressed in DDDs/100 admissions showed an increase, while national hospital consumption expressed in DDDs/100 admissions showed a borderline statistically significant decrease. This decrease was caused by the decrease in antibiotic consumption in other specialized hospitals (rehabilitation, psychiatric, gynaecology and obstetrics) and departments. With the exception of the surgical department, which demonstrated an increase in both DDDs/100 bed days and DDDs/100 admissions (Pearson’s r 0.967 and 0.700, respectively), other departments exhibited diverse trends based on both metrics.

#### Influence of National Audits on Antibiotic Consumption

The influence of national audits on antibiotic consumption is shown in Figure 3. Consumption of J01 one year after an audit, as expressed in DDDs/100 bed days and DDDs/100 admissions, decreased in 9 out of 15 (59.9%) and 11 out of 15 (73.3%) hospitals, respectively. Consumption of antibiotics three or four years after an audit decreased in both DDDs/100 bed days and DDDs/100 admissions in 5 out of 12 hospitals (41.6%). 

A significant disparity was noted in the impact of an audit between general and specialized hospitals. In specialized hospitals, there was a general decrease in antibiotic consumption, as indicated by both metrics. However, the cumulative trend reached significance only in the case of DDDs/100 admissions, reflecting an average decrease of 13.8% (*p* = 0.042). Conversely, in general hospitals, a predominant increase in antibiotic consumption was observed four years post-audit, with a significant cumulative increase noted for both DDDs/100 bed days and DDDs/100 admissions (averaging 5.5% and 5.7%, *p* = 0.027 and 0.018, respectively).

Notably, throughout the observed period, specialized hospitals reported significantly lower DDDs/100 bed days (*p* < 0.001) but higher DDDs/100 admissions (*p* < 0.001) compared to general hospitals.

In all the hospitals, we found some inadequacies and negative results recorded in the national audits. The negative findings recorded at audits in Slovenian hospitals for the period 2013 to 2018 (n = 15) included the following: indications are not documented in the clinical notes; duration of therapy is too long; antibiotics are not prescribed in accordance with local guidelines; critically important antibiotics (CIA) are inappropriately prescribed; there is inappropriate surgical prophylaxis, mostly in terms of duration; consumption data are not reported to health authorities or discussed; appropriate microbiology specimens are not sent, especially in ICUs; inappropriate combinations of antibiotics are used; de-escalation and switching from intravenous to oral therapy is underused; and there are no plans for improving the areas examined in the audit.

## 3. Discussion

This study examined the differences at the national and department levels with regard to the consumption of antibiotics for systemic use during a 14-year period and found that national audits had a greater impact one year after they were conducted compared to three or four years later. In the studied period, the national consumption of antibiotics in hospitals expressed in DIDs (1.52 vs. 1.51) and DDDs/100 admissions (308.1 vs. 298.6) decreased by up to 3.4%, while the consumption expressed in DDDs/100 bed days (43.4 vs. 49.4) increased by almost 14%. The higher consumption of J01 expressed in DDDs/100 bed days can be explained by the 15.5% shorter average LTS. A shorter LOS presented in all departments varies among departments (ranging from 2.7% to 70.1%) and is associated with the more intensive use of antibiotics, which may have negative consequences in terms of antimicrobial resistance. The differences between the lowest and highest yearly consumption expressed in DIDs, DDDs/100 bed days and DDDs/100 admissions were 9.3%, 15.1% and 9.5%, respectively. In 2019, the average consumption of J01 in hospitals in the EU/EEA countries was 1.8 DID (country range: 0.8–2.5), while in Slovenia, it was 17% lower than this. Between 2010 and 2019, no statistically significant changes were observed overall in the EU/EEA countries [15]. A decrease in hospital consumption is more difficult to achieve than a decrease in outpatient antibiotic use due to the greater number of physicians, larger number of different specialists and a broader spectrum of infectious diseases within hospitals [16,17]. In the period from 2006 to 2019, outpatient antibiotic use in Slovenia decreased by 7.7% (12.44 DID vs. 11.48 DID), which is substantially more than the decrease seen in hospitals (by 0.7%). A sustained multifaced approach in outpatients in Slovenia was associated with an overall decrease of antibiotics of J01 by 31% from 1999 to 2012 [18]. Slovenia was not able to reduce J01 consumption in hospital care by 10% and community consumption by 20% in 2022 vs. 2016 (1.49 DID vs. 1.49 DID), as planned in 2016 [19]. While fully guided prescription of antibiotics by infectious disease specialists and face-to-face communication have proved to be effective and safe in this context, this approach is costly and labour-intensive [20,21]. In Dutch hospitals, antimicrobial stewardship teams, consisting of at least an infectious disease specialist, a medical microbiologist and a hospital pharmacist, are responsible for implementing antimicrobial stewardship programmes (ASP). If all activities are performed at a minimal base (one stewardship objective; minimal staffing standard), 0.87 (300 beds) to 1.68 FTE (1200 beds), with a further increase to minimally 1.25 to 3.18 FTE in the following years, with three stewardship objectives monitored (optimal staffing standards during the first few years of implementing an ASP) [22]. Dedicated and sustainable funding for AST is urgently needed to implement comprehensive and functional AST programmes in all healthcare facilities [23]. 

A larger increase in the consumption of J01 was observed at the department level. The consumption expressed in DDDs/100 bed days increased in all departments, ranging from 19.9% to 33.1%, except in medical and surgical ICUs, where it decreased by 7.1% and 5.4%, respectively. The consumption expressed in DDDs/100 admissions decreased only in paediatric departments, by 12.7%, and in the other departments it increased from 10.3% to 39.4%. Increased antibiotic use in DDDs/100 bed days together with a decrease in the length of stay (LOS) by 36.1 to 70.1% could occur with unchanged antibiotic use measured in DDDs per 100 admissions. In our case, antibiotic use measured in DDDs per 100 admissions increased as well, pointing to higher antibiotic prescribing in general. In all departments and in all years, antibiotic use varied widely among departments of the same type. Antibiotic consumption usually differed by a factor of 3 between the highest- and lowest-consuming departments. The gaps between the high and low antibiotic-consuming hospitals and departments have been reported previously [2,10,14]. A significant correlation between consumption expressed in DDDs/100 admissions and case mix index/CMI) (*p* = 0.028) and DDDs/100 bed days and CMI (*p* = 0.008) was found in our previously published study, and this may explain the variations found in the current work [24]. The high variation across hospitals and departments should be further investigated. 

Audit and feedback is a strategy intended to encourage professionals to change their clinical practice. An audit is a systematic review of professional performance based on explicit criteria or standards. This information is subsequently fed back to professionals in a structured manner [25]. Audit and feedback generally lead to small but potentially important improvements in professional practice. The effectiveness of audit and feedback seems to depend on baseline performance and how the feedback is provided [26]. In a French study, the most common subcategories of enablement were audit and feedback [26]. In a recent study, the authors found that audit and feedback are a feasible and valuable methodology to assess the nationwide implementation of antibiotic stewardship programmes in hospitals [27]. Furthermore, it can identify relevant targets for the improvement of such programmes [28]. In our analysis of audits, we uncovered many negative findings, which have been noted in earlier studies [29,30]. We found that there was more likely to be a decrease in antibiotic consumption one year after an audit compared to three or four years later. These data suggest that the effects of audits can be temporary and that audits with feedback should be carried out more often. Moreover, there is a need for greater adherence to the recommendations for improving the prudent use of antimicrobials that are prepared by the audit team.

The limitations of this study include the low number of audits that were analysed. We wanted to include a control group, but the number of general and specialized hospitals was too low to achieve this. However, since the same negative findings were observed in all the hospitals, it is unlikely that analysing more audits would have produced any additional findings.

## 4. Materials and Methods

### 4.1. Population and Hospital Data

Slovenia is a small Central European country, with just 2,108,977 inhabitants, according to the census of January 2021 [31]. Almost all of them (>99%) are covered by compulsory health insurance.

Slovenia has 2 university hospitals, 10 general hospitals and 17 specialized hospitals (rehabilitation, psychiatric, gynaecology/obstetrics, orthopaedics oncology, and hospitals for respiratory and allergic diseases). In total, 26 hospitals are public and 3 are private. Infectious disease (ID) specialists are responsible for implementing antimicrobial stewardship in both university medical centres and in three general hospitals. Two general hospitals get some help from retired ID specialists, and clinical pharmacists and medical microbiologists manage ASP elsewhere. 

University and general hospitals have different departments, including medical (internal medicine), surgery, ICUs (medical, surgery, mixed), paediatrics and gynaecology/obstetrics departments. 

### 4.2. Hospital Antibiotic Consumption Data

Antibiotic consumption for systemic use (J01) data from all hospitals from 2004–2019 were collected using the Anatomical Therapeutic Classification (ATC)/DDDs (WHO version 2019) [32]. At the beginning of this period (2004–2010), we participated in a European research project (ESAC, now ESAC-Net, the European Surveillance of Antimicrobial Consumption Network). Later, hospital surveillance of antimicrobial consumption was a national project, and afterwards national hospital-based surveillance became compulsory as part of a national strategy to combat antimicrobial resistance. We collected national data and data from all the hospitals separately, as well as date from five departments at each hospital. Total hospital consumption was expressed in the number of defined daily doses (DDDs) per 1000 inhabitants per day (DID), number of DDDs/100 bed days and number of DDDs/100 admissions. Departmental consumption data since 2006 were collected from university and general hospitals. The consumption data for the five departments were expressed as the number of DDDs/100 bed days and number of DDDs/100 admissions. Hospital consumption data were provided by hospital pharmacists using ward dispensing records as the source data. The National Institute of Public Health provided national and total hospital data on the number of bed days and number of hospital admissions, while hospital pharmacists provided the department data.

### 4.3. National Activities

The national activities aimed at the rational prescribing of antibiotics in Slovenia before 2016 were published previously [18,33]. European Antibiotic Awareness Day (EAAD) activities have been carried out since 2008, and World AMR Awareness Week activities since 2015, in addition to the activities that continued from previous years (Table 2). 

Audits were performed in hospitals with high consumption of antibiotics for systemic use. The committee of three members looked at the characteristics of the antimicrobial stewardship programme in the hospital and reviewed at least 30 medical charts of patients who received antimicrobials for therapy or prophylaxis. In all hospitals, the medical charts of patients hospitalized in ICUs, medical, surgery, gynaecology/obstetrics and paediatric departments were reviewed. Since 2016, the recommendations have been published online [36].

### 4.4. Statistical Analysis

Statistical analysis was performed in R (R Core Team 2022) [37], and Pearson’s correlation was implemented to test for time-dependent trends in measurements. The efficacy of auditing was assessed by comparing the four-year averages using the Wilcoxon signed-rank test. A threshold of 0.05 was considered statistically significant.

## 5. Conclusions

The study showed that the trends in national antibiotic consumption are different from department-level trends and that the measurement used influenced the trends. Department-level antibiotic consumption data are needed to plan antibiotic stewardship programmes in hospitals. A fall in the consumption of J01 in hospitals is much more difficult to achieve than in an ambulatory care context. We believe that the easiest way to decrease antibiotic consumption in hospitals is to shorten the duration of antibiotic therapy and improve antibiotic surgical prophylaxis [29,38].

## Figures and Tables

**Figure 1 antibiotics-13-00498-f001:**
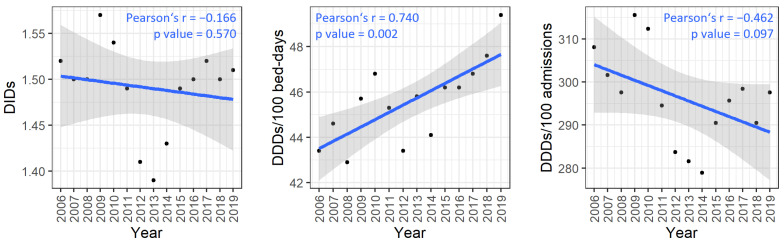
The consumption of antibiotics for systemic use (J01) across all Slovenian hospitals (n = 29) from 2006 to 2019. The trends shown are supported by Pearson’s correlation analysis. A positive Pearson’s coefficient denotes an increase over the observed period, while a negative coefficient indicates a decrease. Abbreviations: DID—defined daily dose per 1000 inhabitants per day; DDD—defined daily dose.

**Figure 2 antibiotics-13-00498-f002:**
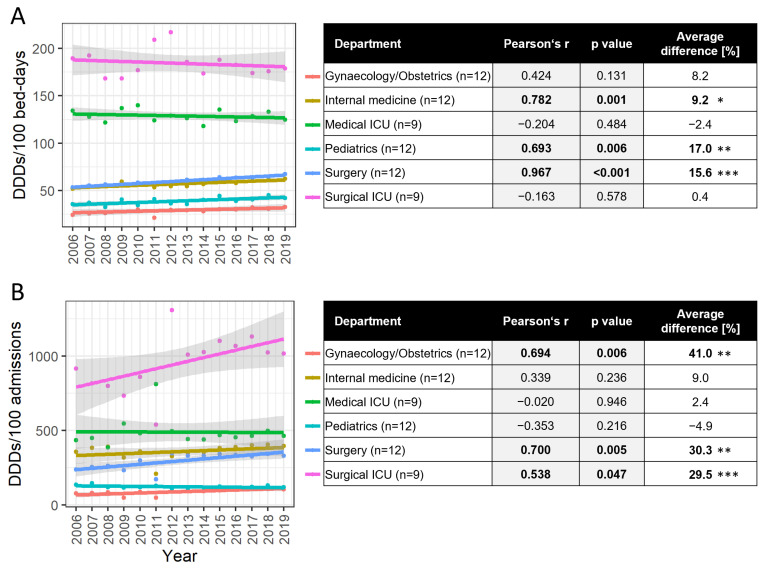
The consumption of antibiotics for systemic use (J01) across various departments. The data are presented in two measures: DDDs/100 bed days (**A**) and DDDs/100 admissions (**B**). Pearson’s correlation analysis was used to assess the trend between 2006 and 2019, with the results presented as Pearson’s r coefficient and corresponding significance (*p* value). A positive Pearson’s coefficient signifies an increase over the observed period, while a negative coefficient indicates a decrease. Additionally, we provide the average difference (%) between the first and last five years within the observed timeframe. Statistical significance is indicated as follows: * *p* = 0.05–0.01, ** *p* = 0.01–0.001, *** *p* < 0.001, with all statistically significant outcomes highlighted in bold.

**Figure 3 antibiotics-13-00498-f003:**
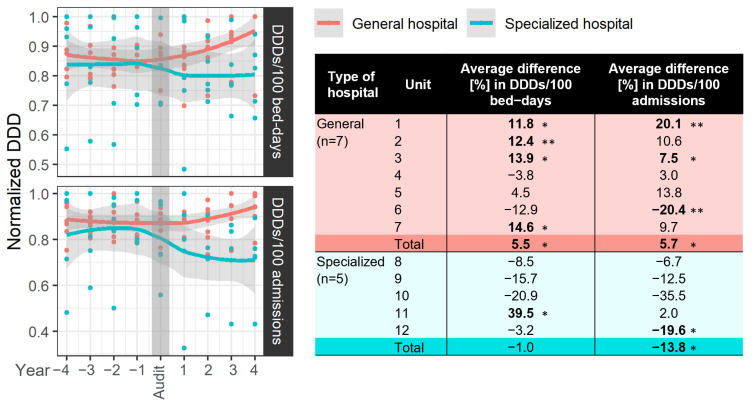
The impact of an audit on antibiotic consumption. To assess this impact, all hospitals were organized chronologically based on the year of the audit, and the average difference (%) was examined within a four-year window before and after the audit. Normalization of absolute values for DDDs/100 bed days and DDDs/100 admissions was performed to enhance the visual representation of the effect. Statistical significance is denoted as follows: * *p* = 0.05–0.01, ** *p* = 0.01–0.001, with all statistically significant outcomes highlighted in bold.

**Table 1 antibiotics-13-00498-t001:** Consumption of antibiotics for systemic use (J01) in all Slovenian hospitals (n = 29) in 2006 and 2019.

Hospital Consumption (J01)	2006	2019	Difference (%) 2006 vs. 2019
DIDs	1.52	1.51	−0.7 *
DDDs/100 bed days	43.36	49.37	+13.8 **
DDDs/100 admissions	308.09	297.63	−3.4 ***
number of beds	9567	9200	−3.9
number of admissions	361,912	387,049	+6.9
number of bed days	2,571,676	2,317,572	−9.9
average length of stay (days)	7.1	6.0	−15.5

DID—defined daily doses per 1000 inhabitants per day; DDD—defined daily doses; * *p* value = 0.570, ** *p* value = 0.002, *** *p* value = 0.097.

**Table 2 antibiotics-13-00498-t002:** National activities in the field of prudent prescription of antibiotics in hospitals in Slovenia.

Activity	Institution (Organizer)	Introduction/Frequency
European Antibiotic Awareness day (EAAD)	Intersectoral Coordination Mechanism (ICM)	2008/every year
World antibiotic Awareness Week (WAAD)	ICM	2015/every year
Article on prudent use of antibiotics	Journal of Medical Chamber	2008/every year except 2015
Article devoted to EAAD	Slovenian Medical Journal	2008 [34]
Conference for the medical directors	ICM	2011/every year
ECDC campaign materials	NIPH	2008/every year
Medical letter	ICM	2008/every year
2-Day symposium	Slovenian Society of Chemotherapy (SSC)	1995/every year
2–3-Day symposium for young specialist	SSC	2010/except 2023
Audits	ICM	2013–2017
Antimicrobial treatment recommendations	Milan Čižman and Bojana Beović	2013 [35]
Antimicrobial treatment recommendations for smart phones	Bojana Beović and Milan Čižman	www.quibaguide.com (accessed on 1 March 2024) [36]

## Data Availability

Data are available in Slovenian language on request.

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
