# Peer review of "The Impact of National Activities on Antibiotic Consumption in Hospitals and Different Departments over a 14-Year Period"

_antibiotics, 2024, doi:10.3390/antibiotics13060498_

Round 1

Reviewer 1 Report

Comments and Suggestions for Authors

Dear Authors,

Congratulations on the excellent manuscript, presented in a very well-structured manner. Antibiotic consumption and resistance are global problems. The manuscript sounds scientific, and the results are clearly presented. Figures and tables are appropriate and clearly present the data. 

Author Response

  Dear Reviewer,

Thank you very much for taking the time to review our manuscript and we are so pleased that your review is positive and you agree with our manuscript.

Reviewer 2 Report

Comments and Suggestions for Authors

Thank you for the opportunity to review this interesting manuscript describing a 14-year review and comparison of hospital antimicrobial consumption. I have some suggestions intended to clarify the presentation of results and conclusions:

1. Please be careful when discussing trends that were not significant. For example, the results portion of the abstract begins by presenting 0.7% and 3.4% decreases in "DDDs" (should this be DIDs??) and DDD/100 admissions, neither of which were significant. Perhaps only the 13.8% increase in DDD/100 bed days (P=0.002) should be mentioned. At the least, I would suggest including the P values for all 3 observations. Accordingly, please include P values in Table 1.

2. The last sentence of the abstract and conclusions state "The easiest way to reduce hospital consumption of antibiotics is to shorten the duration of antibiotic therapy and improve surgical prophylaxis", yet the study results and discussion do not discuss these points in detail. Do you have the data to present and elaborate on these extremely important issue in more detail?

(2a) Can you present the average duration of antibiotic use for both 2006 and 2019, both overall and by department (in particular the Department of Surgery and the Surgical ICU)?

(2b) Figure 2 shows us that the Surgery Department accounted for 30.3% and 15.6% increases in DDD/100 admissions and DDD/100 bed days, respectively, while the Surgical ICU accounted for a 29.5% increase in DDD/100 admissions, but no change in DDD/100 bed days. Is the increase in DDD/100 admissions for both units a reflection of 6.9% increase in overall admissions shown in Table 1? Can you also present the change in surgical admissions?

(2c) As I read the manuscript, I associate the Department of Surgery with surgical prep on admission, and the initiation of surgical prophylaxis on call to OR, whereas in the Surgical ICU I think about stopping surgical prophylaxis appropriately post-op, hopefully followed by patient recovery and discharge. Table 1 shows a 15.5% decrease in overall average length of stay--how does this relate to the above metrics with respect to DDD/100 bed days? Did prudent and appropriate surgical prophylaxis contribute to a stable average length of stay in the Surgical ICU, reflected by only a 0.4% change in DDD/100 bed days? 

Author Response

Dear Reviewer,

Thank you very much for taking the time to review our manuscript.

We are grateful for your suggestions for improvement and we tried to amend the text accordingly.

  1. Please be careful when discussing trends that were not significant. For example, the results portion of the abstract begins by presenting 0.7% and 3.4% decreases in "DDDs" (should this be DIDs??) and DDD/100 admissions, neither of which were significant. Perhaps only the 13.8% increase in DDD/100 bed days (P=0.002) should be mentioned. At the least, I would suggest including the P values for all 3 observations. Accordingly, please include P values in Table 1.

Reply: We deleted the trends expressed in DID and DDD/100 admissions and    mentioned only significant values. We included P values in the Table 1 as well.

  1. The last sentence of the abstract and conclusions state "The easiest way to reduce hospital consumption of antibiotics is to shorten the duration of antibiotic therapy and improve surgical prophylaxis", yet the study results and discussion do not discuss these points in detail. Do you have the data to present and elaborate on these extremely important issue in more detail?

Reply: We deleted  the last sentence in the abstract.  

(2a) Can you present the average duration of antibiotic use for both 2006 and 2019, both overall and by department (in particular the Department of Surgery and the Surgical ICU)?

Reply: We cannot present the average duration of antibiotic use, we collected aggregated data only. We can present the average LOS of stay in different departments in 2006 and 2019.

(2b) Figure 2 shows us that the Surgery Department accounted for 30.3% and 15.6% increases in DDD/100 admissions and DDD/100 bed days, respectively, while the Surgical ICU accounted for a 29.5% increase in DDD/100 admissions, but no change in DDD/100 bed days. Is the increase in DDD/100 admissions for both units a reflection of 6.9% increase in overall admissions shown in Table 1? Can you also present the change in surgical admissions?

Yes, DDD/100 admissions is a reflection of 6,9% increase in overall admissions.. . Number of admissions increased in surgery by 32,4% and in surgical ICUs departments by 30,2% respectively.

(2c) As I read the manuscript, I associate the Department of Surgery with surgical prep on admission, and the initiation of surgical prophylaxis on call to OR, whereas in the Surgical ICU I think about stopping surgical prophylaxis appropriately post-op, hopefully followed by patient recovery and discharge. Table 1 shows a 15.5% decrease in overall average length of stay--how does this relate to the above metrics with respect to DDD/100 bed days? Did prudent and appropriate surgical prophylaxis contribute to a stable average length of stay in the Surgical ICU, reflected by only a 0.4% change in DDD/100 bed days? 

Reply: A decrease in LOS coupled with stable DDDs results in an increase in DDD/100 bed-days. Since DDD/100 bed-days only increased by 0.4%, it reflects lower use of antibiotics per patient. For the purpose of this study we have not collected data on appropriateness of surgical prophylaxis. The last ECDC point-prevalence survey on antibiotic use in 2016-2017 revealed that in Slovenian acute care hospitals approx. 50% patients still receive surgical prophylaxis for more than one day, which is not an ideal situation. We could speculate that overall antibiotic use improved which contributed to better patient care and shorter ICU LOS.  

European Centre for Disease Prevention and Control. Point prevalence Survey of healthcare-associated infections and antimicrobial use in European acute care hospitals. 2016-2017. Stockholm:,ECDC;2017

Reviewer 3 Report

Comments and Suggestions for Authors

Dear Authors,

Overall, a well written and interesting study. I have highlighted a few bits of the text with comments on the left side.

The statement in your abstract: The easiest way to reduce hospital consumption of antibiotics is to shorten the duration of antibiotic therapy and improve antibiotic surgical prophylaxis borders on being bizarre. Hospital consumption of antibiotics has to be led by other factors, such as the microbiology of infection, the minimum duration of use of antibiotics based on the product SmPC, treatment guidelines and various other factors. 

Please clarify your abbreviations of DDD and DID in the text. You introduced DID as defined daily doses per 1,000 population per day, yet you did not introduce DDD, and they read almost as if they are interchangeable (I know what you mean, but best to clarify early on, not only under the figures).

Section 4.3 is informative but does not really flow as part of the method. Perhaps this would be better in tabular form, easier for the reader to see events chronologically.

Inappropriate antibiotic use is more likely in the community where the expertise of a microbiologist is not used. Perhaps it's worth mentioning this difference between the 2 settings.

You jumped from the Introduction to the Results. In order to appreciate what the results imply, I would prefer to first read the methodology, then once I have understood that, see the results. I suggest that you switch this around.

Author Response

Dear Reviewer,

Thank you very much for taking the time to review our manuscript. We are grateful for your suggestions for improvement and we tried to amend the text accordingly. 

  1. The statement in your abstract: The easiest way to reduce hospital consumption of antibiotics is to shorten the duration of antibiotic therapy and improve antibiotic surgical prophylaxis borders on being bizarre. Hospital consumption of antibiotics has to be led by other factors, such as the microbiology of infection, the minimum duration of use of antibiotics based on the product SmPC, treatment guidelines and various other factors. 

Reply: The sentence “The easiest way to reduce hospital consumption of antibiotics is to shorten the duration of antibiotic therapy and improve antibiotic surgical prophylaxis” was deleted.

  1. Please clarify your abbreviations of DDD and DID in the text. You introduced DID as defined daily doses per 1,000 population per day, yet you did not introduce DDD, and they read almost as if they are interchangeable (I know what you mean, but best to clarify early on, not only under the figures).

Reply: In the Chapter Materials and methods and in abstract we clarified the abbreviation DDD

  1. Section 4.3 is informative but does not really flow as part of the method. Perhaps this would be better in tabular form, easier for the reader to see events chronologically.

Reply: We included the tabular form.

Table 2.National activities in the field of prudent prescrebing of antibiotics in hospitals in Slovenia

Activity

Institution (organizer)

Introduction/frequency

European Antibiotic Awareness day (EAAD)

Intersectoral Coordination Mechanism (ICM)

2008/every year

World antibiotic Awareness Week (WAAD)

ICM

2015/every year

Article on prudent use of antibiotics

Journal of Medical Chamber

2008/every year except 2015

Article devoted to EAAD

Slovenian Medical Journal

2008

Conference for the medical directors

ICM

2011/every year

ECDC campaign materials

NIPH

2008/every year

Medical letter

ICM

2008/every year

2-day symposium

Slovenian Society of Chemotherapy (SSC)

1995/every year

3-day courses for young doctors

SSC

2010/except 2023

Audits

ICM

2013-2017

Antimicrobial treatment recommendation booklet

M.C in B.B.

2013

Antimicrobial treatment recommendations as smart-phone application

B.B. and M.C.

www.quibaguide.com

  1. Inappropriate antibiotic use is more likely in the community where the expertise of a microbiologist is not used. Perhaps it's worth mentioning this difference between the 2 settings.

Reply: Our experience is different. As already mentioned in the paper, in Slovenia outpatients antibiotic use decreased by 31% expressed in DIDs from 1999 to 2012. Multifaceted measures were introduced (32).

  1. You jumped from the Introduction to the Results. In order to appreciate what the results imply, I would prefer to first read the methodology, then once I have understood that, see the results. I suggest that you switch this around.

Reply: We do agree with you but the order of the chapters is required by Instructions for authors in the Research Manuscript section.

Round 2

Reviewer 2 Report

Comments and Suggestions for Authors

Thank you very much for addressing my comments. I fully agree with all of your responses and changes. This is an outstanding manuscript that addresses a critical worldwide issue. Best of luck moving forward!